# Effects of Digital Learning and Virtual Reality in Port-A Catheter Training Course for Oncology Nurses: A Mixed-Methods Study

**DOI:** 10.3390/healthcare11071017

**Published:** 2023-04-03

**Authors:** Shu-Feng Shih, Li-Ling Hsu, Suh-Ing Hsieh

**Affiliations:** 1Department of Nursing, Koo Foundation Sun Yat-Sen Cancer Center, Taipei City 11259, Taiwan; z650417@kfsyscc.org; 2Department of Nursing, Ching Kuo Institute of Management and Health, Keelung 203301, Taiwan; llhsu3684@gmail.com; 3Department of Nursing, Chang Gung University of Science and Technology, Taoyuan City 33303, Taiwan

**Keywords:** digital learning, virtual reality, implanted Port-A catheter, nurses, knowledge, skills, learning attitude, satisfaction

## Abstract

In-service education for oncology nurses usually adopts didactic teaching. This study investigated the effects of virtual reality (VR) and a digital learning-based Port-A-catheter educational course for oncology nurses. A mixed-methods research design was employed, with a convenience sample of 43 nurses from a regional teaching hospital in Taiwan participating. Measurements were taken at three time points: pre-test, 1st post-test, and 2nd post-test. The data was analyzed using descriptive statistics and repeated ANOVA tests. Results showed significant improvement in Port-A-catheter knowledge and skill levels (*p* < 0.0001) and high learning attitude and satisfaction scores of 4.29 ± 0.46 and 4.31 ± 0.58 points, respectively. Five qualitative themes emerged, highlighting the realistic VR scenarios, VR practice’s usefulness, willingness to learn with VR, VR system limitations, and the potential for future courses. The study concluded that a VR-based educational course effectively enhanced nurses’ knowledge, skills, learning attitude, and satisfaction, recommending the inclusion of diverse clinical scenarios for practical learning.

## 1. Introduction

Cancer has been the leading cause of death in Taiwan for the last 40 years, accounting for 28.0% of total deaths [1]. In 2020, there were 121,979 new cancer cases in Taiwan, with an incidence of 517.71 per 100,000 person-years [2]. Globally, 19.3 million people were diagnosed with cancer in 2020, and cancer accounted for 9.96 million deaths, according to the International Agency for Research on Cancer (IRAC) [3]. Chemotherapy, with or without targeted therapy, is used to improve cancer patients survival rates alongside surgery and radiotherapy [4]. Implantable central venous access (Port-A catheter) is a common method used for long-term infusion of drugs for chemotherapy, nutrition, and blood transfusion in cancer patients. However, complications such as puncture failure, infection, catheter fracture, and extravasation can occur, potentially resulting in patient death. Good puncture technique and care are critical for catheter maintenance [5,6,7,8]. Injection through a Port-A catheter is an invasive procedure, and a good puncture greatly eliminates the damage to the catheter path [9,10]. Professional knowledge of Port-A management is essential for nursing staff to improve cancer care quality [6].

However, inadequate Port-A catheter education and training can lead to clinical mistakes that endanger patient lives. Standardized operating procedures and thorough education and training can improve nursing staff’s attitude and motivation toward Port-A catheter care [11,12]. Few hospitals in Taiwan include Port-A catheter management in their on-the-job training for new employees, and only a 2 h course is offered in the 2–3 basic cancer nursing training courses organized by the Taiwan Oncology Nursing Society [13,14,15,16,17].

With advancements in computer technology, virtual reality (VR) is increasingly used in training courses and has the greatest potential for course development [18]. There are three characteristics of a successful VR experience: interaction, immersion, and imagination [19]. VR simulations in nursing education enhance professional knowledge, clinical reasoning skills, knowledge retention, and learning satisfaction [20,21]. VR experiences align with modern learners’ expectations and learning styles [18,21,22].

However, only three studies have investigated the use of VR in nursing education Port-A catheter management. Two quasi-experimental studies and one meta-analysis were conducted. Tsai et al. [23] developed a VR simulation system and compared the knowledge, skills, satisfaction, and frequency of incorrect selection and operating time in Port-A catheter management between novice nursing staff with or without experience in the VR simulation system. They found significant improvements in knowledge, total time, total error rate of equipment, and other factors in the experimental group. Jung and Park [24] developed a VR headset-based nursing course for Chemoport insertion surgery and evaluated the knowledge, learning attitude, satisfaction with self-practice, and learning motivation. The experimental group showed significant improvements in knowledge, learning attitude, and satisfaction with self-practice. Chen et al. [20] conducted a meta-analysis of 12 RCT or quasi-experimental studies to investigate the effectiveness of VR-based nursing education on knowledge, skills, satisfaction, confidence, and performance time. They found significant improvements in knowledge but no intergroup differences in other factors. In summary, there is limited research on VR-based nursing education for Port-A catheter management. This study aims to examine the effects of a digital learning and VR-based educational course on oncology nurses’ knowledge, skills, learning attitude, and learning satisfaction with Port-A catheter management.

This study is based on situated learning theory, which holds that the situation is essential for learning to occur and progress. Clues embedded in the situation help learners unintentionally remember external events, facilitating memory retention and retrieval. Learners with similar past experiences or situations quickly immerse themselves in the learning process, significantly aiding clinical judgments [25,26]. Opportunistically provided clues in the scenario can enhance deeper and more active learning of real clinical situations, fostering learners’ inferential capability, problem solving, clinical decision-making, and reflection [27].

To motivate nurse practitioners to learn Port-A catheter care and become familiar with related skills, a VR simulation of a real implantable Port-A catheter operation was integrated into the course to facilitate the identification of key points in Port-A catheter nursing care. A fully immersive VR situational learning experience was incorporated through a computer system, VR headsets, and remote controllers. This approach aims to provide a considerate and interactive Port-A catheter injection experience in the virtual world, offering a safe, friendly, and realistic learning environment without time or practice limitations. This method helps motivate and facilitate learning, improve learning outcomes, and apply knowledge and skills in the clinical care of Port-A catheter injections.

## 2. Materials and Methods

### 2.1. Study Design

This study employed a mixed-methods research design, utilizing a quasi-experimental single-group repeated measures design to teach “Management and Prevention of Abnormalities in Implanted Port-A Care” through the Training Management System (TMS) digital program. The “Implanted Port-A Catheter Care Virtual Reality Teaching System” served as the intervention tool. The pre-test (knowledge and DOPS), post-test within one week of intervention completion (knowledge, DOPS, learning attitude, satisfaction), and delayed post-test one month after intervention completion (knowledge and DOPS) were conducted following Tsai et al. [23] and Ekstrand et al. [28], taking into account factors such as participant numbers, clinical practice, and time constraints.

Additionally, 10 voluntary participants were interviewed one week after the intervention using a semi-structured format, allowing free responses. Interviews lasted about 20–30 min, and upon completion, a verbatim transcript was produced. The e-transcript content was repeatedly read and compared to identify relationships and meanings in the obtained information. Interview data were coded, categorized, and analyzed to form themes and classifications, offering deeper insight into the intervention’s effectiveness. (Table 1).

### 2.2. Participants and Setting

This study used convenience sampling to recruit nursing staff who implemented Port-A catheter care and had been working for up to 2 years at a regional teaching hospital in Northern Taiwan. Exclusion criteria included unwillingness to participate, incomplete research process participation, left-handedness, and not completing the pre- and post-tests. Left-handed learners, who may require specific interfaces and control methods, were excluded due to increased design difficulty and cost. As no similar studies were available for effect size estimation and pilot studies were not feasible, the required sample size was calculated using G*power 3.1 for paired t-tests with an effect size of 0.5, α = 0.05, and β = 0.8 [29], resulting in an estimated 34 participants. Accounting for a 10% of sample loss, the sample size was adjusted to 38 participants. The observed powers for knowledge and skills in within-subjects effects of repeated ANOVA measures were 0.998 and 1.000, respectively.

### 2.3. Intervention

#### 2.3.1. VR Intervention of a Real-Situated Teaching System on Implanted Port-A Catheter Care

##### Development of the Virtual Reality Teaching System

The virtual reality teaching system in this study was developed after multiple discussions and revisions with three programmers and five nursing experts. The system was based on nursing care and technical procedures from the Koo Foundation Sun Yat-Sen Cancer Center (KFSYSCC) [30] and a literature review of domestic and international journals. The system covers the entire Port-A catheter care procedure, including operating instructions and potential situations encountered during the process. The VR simulation program focuses on integrating real clinical steps into the scenario.

To ensure understanding and proficiency, 21 checkpoints were defined within the scenario. After completion, three clinical nursing and education experts evaluated the content’s accuracy and suitability. Both Scale-Content Validity Index (S-CVI) and Item-CVI were 1.00, and revisions were made based on expert recommendations. The teaching system is divided into three parts: (1) preparation of the Port-A catheter before injection, including checkpoints such as checking the doctor’s orders, explaining the purpose, handwashing, evaluation, and material preparation; (2) Port-A catheter injection process steps and related precautions, including checkpoints like checking the patient, finding the optimal insertion site, disinfection, sterile placement of items in the kidney, curved needle insertion, and back-drawing without blood return; and (3) mistake records. After completing the Port-A catheter simulation, the system provides records of mistakes, error causes, and correct operations, allowing learners to better understand their performance and areas for improvement. Figure 1 illustrates the Port-A catheter implantation in the VR teaching system. The content management system (CMS) interface includes user account and password settings, and clinical nursing teachers can view each learner’s completion rate, practice and test times, video-watching records, and more as supplementary evaluation tools.

The goal of developing the virtual reality teaching system is to enhance new nurses’ ability to perform correct Port-A catheter implantation procedures and foster a proactive attitude towards learning and practical application among nursing staff. The VR simulation of Port-A catheter implantation features multiple feedback processes, including text messages, voice messages, handle vibration, auxiliary views, and records of mistakes. The records of mistakes, displayed on-screen as text or images, show the cause of the mistake and instructions for the correct operation, helping learners catch details they may miss or neglect during daily operations. Figure 2 presents the 3D VR learning system, and Figure 3 displays the operation of the VR teaching system.

The virtual reality teaching system provides standardized teaching information in a stimulating environment and personalized feedback-based trainee learning progression during the training process. To ensure consistency and standardization across all involved units, the system was set up and configured individually by the authors. The authors also addressed any operational or learner issues to ensure consistent training and operation. This approach aims to ensure that all learners receive the same level of training and minimize potential discrepancies due to unit-specific training settings.

###### Digital Course of “Implanted Port-A Abnormality Management and Preventive Care”

The digital course is divided into two parts. The first part covers the management of Port-A catheter abnormalities and preventive care, while the second part provides instruction on operating the VR simulation of Port-A catheter care. During the research process, to prevent misinterpretation of results due to users’ unfamiliarity with the VR operating system, a 5 min demonstration video recorded by the research staff was played at the end of the course, accompanied by written instructions, to facilitate post-course practice. The teaching program’s design is shown in Table 2.

#### 2.3.2. Interventional Procedures

According to the course arrangements, participants received the digital “Implanted Port-A abnormality management and preventive care” course from 11 January 2018 to 15 January 2018. They carried out the VR-simulated Port-A catheter care intervention unit by unit from 15 January to 9 March 2018. Each unit spent five days practicing the VR simulation, with research staff available to assist at any time. Participants were free to contact the research staff via text message, video communication by phone, or the LINE messaging app for immediate problem solving. No participants withdrew until the end of the study. The intervention and arrangement of the participation in each section are illustrated in Table 3.

### 2.4. Instruments

A background information questionnaire, a Port-A DOPS evaluation form, a knowledge scale for a virtual reality training system for an implanted Port-A catheter, a learning attitude scale, and a satisfaction questionnaire are among the instruments used in this study. Paper and pencil tests are used for all instruments.

#### 2.4.1. Background Information Questionnaire

The background information questionnaire, based on Martin and Ertzberger [31], investigates information and experiences about mobile learning among college students. Items include gender, age, education, professional nursing licenses, on-the-job training, seniority in the hospital and nursing, advanced level of care, work unit, and access to information about VR system devices and VR experiences.

#### 2.4.2. Port-A DOPS Evaluation Form

Adopting the Port-A Direct Observation of Procedural Skills (DOPS) rating form used in KFSYSCC, which is based on domestic and international literature [32,33], the form evaluates nursing staff’s implementation of Port-A skills with 11 items. These items include understanding the technique and procedure, informed consent, appropriate analgesia and safe sedation, preparation of materials, technical ability to perform skills safely, seeking help when appropriate, aseptic technique, post-procedure management, communication skills, consideration of the patient/professionalism, and overall performance. Trainee performance was rated as “below the standard” (1, 2 points), “close to the standard” (3 points), “meeting the standard” (4, 5 points), “advanced” (6 points), or not applicable (NA). Higher scores indicate better Port-A technique. After deleting items 4 and 7 due to low item usage frequency and correcting item-total correlation values < 0.30 at the three time points, the Cronbach’s α for this sample ranged from 0.90 to 0.97. The average intraclass correlation coefficient for test-retest reliability across three time points was 0.73 (*p* < 0.0001). The principal component analysis with promax rotation showed one factor, explaining 54.22–81.02% of variances across the three time points.

#### 2.4.3. Virtual Reality Teaching System Knowledge Scale of Implanted Port-A Catheter

Based on “Port-A catheter Nursing and Management of Abnormal Situation” from KFSYSCC and analyzed according to Bloom’s taxonomy of cognitive objectives [34], this scale evaluates learners’ knowledge and effectiveness of learning of Port-A catheter technique and nursing process, identifying Port-A catheter problems by signs and symptoms, common abnormal situations in Port-A catheter technique, and clinical decision-making steps. After review by five experts, two items were removed from the original 22-item knowledge scale based on expert opinions. One item was removed because its I-CVI was less than 0.80, and another because its answer could be correctly deduced from previous items. Furthermore, seven items were removed due to low discriminant validity. The final 13-item multiple-choice knowledge scale has an S-CVI of 0.85 and an I-CVI of 0.97. Correct answers score 1 point, while incorrect answers score 0 points. Total possible scores range from 0 to 13 points, with higher scores indicating better knowledge of the Port-A catheter. The difficulty rating ranged from 0.73 to 0.86, and the discrimination ranged from 0.28 to 0.42 for this sample at three-time assessment points. The average intraclass correlation coefficient for test-retest reliability across the three time points was 0.05 (*p* < 0.0001).

#### 2.4.4. Virtual Reality Teaching System: Learning Attitude Scale for Implanted Port-A Catheter

Referring to Martin and Ertzberger’s [31] mobile Learning Attitude Scale, this scale assesses nurses’ attitudes toward the virtual reality teaching system for implanted Port-A catheter care, perceived ease of use, usefulness, learning motivation, content importance, and subjective cognition and perception. The original 15-item scale included two open questions. After expert review, three items were removed, leaving 12 items with an S-CVI and I-CVI of 1. The 5-point Likert scale ranges from 5 points for strongly agreeing to 1 point for strongly disagreeing, with possible total scores ranging from 12 to 60 points. Higher scores indicate a more positive attitude toward learning. The Cronbach’s internal consistency reliability was 0.89 after one post-test with the study population. The principal component analysis with promax rotation showed three factors, explaining 67.28% of variances.

#### 2.4.5. Virtual Reality Teaching System Satisfaction Questionnaire for Implanted Port-A Catheter

Referring to Yu et al. [35], this questionnaire assesses participants’ satisfaction and positive feelings towards the VR teaching system for an implanted Port-A catheter [36]. It includes six closed items and two open items, using a 5-point Likert scale ranging from 5 points for strongly agreeing to 1 point for strongly disagreeing. Total possible scores range from 6 to 30 points, with higher scores indicating better satisfaction with the VR teaching system. After review by five experts, both S-CVI and I-CVI were 1, indicating good content validity. The Cronbach’s internal consistency reliability was 0.92 after one post-test with the study population. The principal component analysis with promax rotation showed one factor explaining 72.02% of variances.

#### 2.4.6. Interview

Trained postgraduates conducted face-to-face interviews with 10 participants, using six semi-structured questions (Table 4) to explore nursing staff’s feelings and experiences during the VR learning process.

### 2.5. Procedure

After obtaining signed informed consent from all participating nursing staff, the research team explained the VR teaching system’s operation for caring for patients with Port-A catheter implantation and provided a video demonstration. The study recruited nursing staff working in a regional teaching hospital in Northern Taiwan (from 1 December 2015 to 30 November 2017) in care units where Port-A catheter implantation skills were required and had less than two years of work experience. The intervention consisted of a course on caring for patients with implanted Port-A catheters and a VR teaching system. Participants completed a pre-test (background information, knowledge, and DOPS) two weeks before the intervention, a 1st post-test (knowledge, DOPS, learning attitude, and satisfaction) within one week after the intervention, and a delayed post-test (knowledge and DOPS) one month after the intervention, as shown in Figure 4.

### 2.6. Data Analysis

Data was analyzed using SPSS 21.0 software (IBM Corporation, Armonk, NY, USA). Normality and outlier assumptions were checked. Descriptive statistics included number, percentage, mean, standard deviation (SD), range, and median. Inferential statistics used repeated-measures ANOVA to assess within-subjects effects with post hoc tests of simple contrasts. *p* values less than 0.05 were considered statistically significant. Qualitative interview data was analyzed using content analysis. Verbatim transcripts were completed within 2–3 weeks after the interviews, and researchers repeatedly reviewed transcripts to compare information and the meanings of words and sentences. The transcripts were coded, categorized, and analyzed, with themes formed and named based on the transcripts.

### 2.7. Ethical Considerations

Ethical approval was obtained from the Institutional Review Board (No. 20170531A). All participants signed informed consent forms. They volunteered to participate in the study and were able to withdraw at any time. Questionnaires were collected anonymously.

## 3. Results

### 3.1. Background Information

A total of 43 females, ranging in age from 20 to 29 years old, participated in the study. The average age was 22.88 years. Most participants were 22 years old (32.6%), followed by 23 years old (27.9%). Regarding education, the majority had two-year college degrees (55.8%), followed by university degrees (18.6%). The average length of employment at the hospital was 9.86 ± 6.27 months, and the average length of total nursing work experience was 12.19 ± 9.6 months. In terms of the nursing clinical ladder, the most common was N1 nurse (90.7%). For virtual reality experience and knowledge, 67.4% of participants were familiar with VR, with television (46.5%) being the most common source of information. In terms of experience using the virtual reality system, 69.8% had never used it before (Table 5).

### 3.2. Use Status of Virtual Reality Teaching Intervention Learning System on Implanted Port-A Catheter Care

All nursing staff attended the accessed VR teaching material for implanted Port-A abnormality management and preventive care on the hospital’s digital learning platform within the open hours. Each participant took approximately 58 min to complete the digital course. In addition, 43 nursing staff practiced VR training at least three times, with a total average training time of approximately from 9 to 12 min. The mean level hint frequency ranged from 0.28 to 1.23 times, and the mean error frequency was between 3.98 and 7.37 times. As the number of practices increased, the number of practitioners decreased, and the total average training time, level hint frequency, and error frequency also decreased with increased practice frequency (Table 6).

### 3.3. Within-Subjects Effects on Knowledge and Skills

#### 3.3.1. Knowledge Test of Implanted Port-A Catheter

The average pre-intervention knowledge test score was 9.53 ± 2.23. Scores increased to 10.79 ± 1.47 points within one week and 10.84 ± 1.46 points within one month post intervention. RM-ANOVA revealed a significant difference (F_(1.55, 65.07)_ = 17.14, *p* < 0.0001, Partial η^2^ = 0.290), and post hoc analysis indicated that the average knowledge scores of the first post-test (F_(1, 42)_ = 25.85, *p* < 0.0001) and the second post-test (F_(1, 42)_ = 17.91, *p* < 0.0001) were significantly higher than the pre-test (Table A1).

#### 3.3.2. Port-A Catheter Curved Needle Injection Skill

The average pre-intervention DOPS was 29.35 ± 6.54, increasing to 36.93 ± 4.66 after one week and 39.26 ± 3.49 after one month. RM-ANOVA revealed a significant difference (F_(1.49, 62.45)_ = 90.45, *p* < 0.0001, Partial η^2^ = 0.683), with post hoc analysis showing that the average DOPS score of the first (F_(1, 42)_ = 65.27, *p* < 0.0001) and second (F_(1, 42)_ = 158.03, *p* < 0.0001) post-tests were significantly higher than the pre-test. The overall satisfaction also significantly improved, with the DOPS assessment being 3.65 ± 0.87 before the intervention, 4.63 ± 0.66 after the first post-test, and 4.91 ± 0.53 after the second post-test. RM-ANOVA showed a statistically significant difference (F_(1.58, 66.18)_ = 94.59, *p* < 0.0001, Partial η^2^ = 0.693), and post hoc analysis showed that the average DOPS score of the first (F_(1, 42)_ = 68.98, *p* < 0.0001) and the second (F_(1, 42)_ = 200.77, *p* < 0.0001) post-tests were significantly higher than the pre-test score (Table A1).

### 3.4. Learning Attitude

The nursing staff’s top scores in learning attitude toward the virtual reality learning system were “The virtual reality teaching system is a realistic and uncluttered scene, and is easy to identify” at 4.49 ± 0.59, “The virtual reality teaching system provides accurate information” at 4.42 ± 0.63, and “The virtual reality teaching system enhances my interest in learning” at 4.40 ± 0.70. Conversely, the bottom scores for learning attitude were “I like to learn with virtual reality (VR) devices” at 4.14 ± 0.74, “I will be interested in using this method to learn in the future” at 4.16 ± 0.75, and “The length of the virtual reality teaching system course is appropriate” at 4.19 ± 0.82. Learning attitude is classified into three levels of satisfaction (disagree, neutral, and agree), and the grand mean of the percent for learning attitude was 88.18% (Table 7).

### 3.5. Satisfaction

Table A2 reports the satisfaction scores, with “Virtual reality teaching system matches real world curriculum” ranked the highest at 4.42 ± 0.63, followed by “Learning process for virtual reality teaching system is sequential and logical” with a score of 4.35 ± 0.75, and “Virtual reality teaching system covers appropriate scopes” with a score of 4.35 ± 0.69. The item with the lowest satisfaction was “The difficulty of virtual reality teaching system is at moderate level” at 4.19 ± 0.70, followed by “I am satisfied with the overall effectiveness in learning the virtual reality teaching system” at 4.23 ± 0.68, and “The process of teaching virtual reality teaching system is clear” with a score of 4.30 ± 0.67. The overall average satisfaction score was 4.31 ± 0.58. Satisfaction with the learning system was categorized into three levels (dissatisfied, common, and satisfied), with an overall satisfaction rate of 90.7%. The grand mean of the percent of satisfaction was 90.32%.

Qualitative feedback revealed that 24/28 nurses provided positive feedback, with most comments relating to the system’s user-friendliness and realism. Negative feedback was provided by four nurses, mainly regarding difficulty in using the system and discomfort while wearing the headset.

### 3.6. Use Experience from an Interview

The 10 interviewees were all female and university-educated, with work experience ranging from 4 months to 2 years and no prior virtual reality-related experience.

The interview data were categorized into five major themes: (1) impressive and realistic virtual reality (VR) scenarios; (2) helpfulness of VR practice; (3) willingness to learn VR in the future; (4) limitations of VR systems; and (5) potential for the planning of future courses. The feedback described for each theme is separated as follows.

#### 3.6.1. Impressive and Realistic Virtual Reality (VR) Scenario

Related subthemes include “deepening the impression of operating procedures”, “more comprehensive learning”, “familiarizing with the technique”, and “repeated practice can reduce the waste of consumables”:

S1: “In the past, we learned from books or slide presentations, but this time we learned through VR. We got immediate feedback such as incorrect or inappropriate operation of particular steps. It’s very impressive.”;

S8: “It reduces the waste of consumables. If we practice like we used to, the repeated opening of sterile consumables made our management of sterile technique less accurate, and the fluency of wearing gloves is not as good as in VR.”;

S10: “The experience of push/stop is quite impressive. It is carried out by pressing a flat disc, which is similar to the amount I push during actual clinical practice. Therefore, I can know whether the amount I push each time is OK or not for tube washing. It helps me to know whether the amount I push during actual clinical practice is OK or not.”

#### 3.6.2. Virtual Reality Practice Is Helpful

Related subthemes include “more careful movements”, “more confident in practical operations”, “self-evaluation of capacities”, “comprehensive understanding of operational procedures”, etc.

S1: “It’s helpful. There were vibration reminders for some movements. I am worried about not going through a certain point, so I am more attentive, more careful, and more precise when performing each movement. I am very afraid of making mistakes.”;

S3: “It’s helpful because it allows me to self-evaluate while self-learning.”;

S6: “It’s helpful, especially for getting familiar with the preparation and the initial treatment when there is no blood return.”

#### 3.6.3. Willing to Learn with Virtual Reality in the Future

Related subthemes include “fresh experiences”, “sense of achievement”, “reducing time and labor”, “self-learning”, etc.

S1: “If there are continued VR courses in the future, I will be willing to learn because passing them successfully will give me a great sense of accomplishment.”;

S2: “I think it’s pretty fun, and the experience is quite fresh. If there are continued VR courses in the future, I will continue to learn.”;

S8: “I am willing to learn because I think this will save a lot of time. The preceptor may not even need to be there. I can keep practicing without any problems.”

#### 3.6.4. Limitations of Virtual Reality Systems

Related subthemes include “lack of reality”, “cannot perform fine motor skills”, “poor system sensing”, “confined spaces”, etc.

S2: “Overall, this system is good, however, there was no real sense while inserting the needle. In addition, the sensation in this system is somehow not friendly for the shorter learners.”;

S5: “I think it’s more accurate to say that this is a limitation of current VR technology rather than a drawback to handle fine motor skills in VR experience. And it still needs to be combined with actual clinical practice.”;

S7: “It’s better to have the VR experience in a wider area, so it is less likely to collide. It is difficult to simulate actual conditions when it comes to fixed needle insertion force.”

#### 3.6.5. Direction for the Planning of Future Curriculum

Related subthemes include “suitable for pre-job training”, “better understanding of the content of the course”, and “appreciating the importance of skills.”:

S6: “I suggest using VR for pre-job training. I can practice repeatedly on my own, which would better prepare me with more complete prior knowledge before I carry out real clinical operations on patients. I will have more confidence, and it will be very helpful.”;

S10: “I think the preceptor can first demonstrate the VR operation during pre-job training, and then let them practice on their own. I think it will be very effective. It will allow new students to focus on learning the technical process, rather than being confused by too much information from the preceptor during pre-job training. After we have mastered the technical skills, the preceptor can guide us on how to deal with other problems in the clinical setting, which I think will be better.”

## 4. Discussion

### 4.1. Learning Theory of Virtual Reality Teaching

This study’s virtual reality teaching system, based on situational learning theory, emphasizes the importance of context in learning and knowledge retention [18,19]. This study shares similarities with Tsai et al. and Jung and Park. Tsai et al. had nurses in the experimental group practice a 40 min Port-A catheter insertion simulation and repeat it within three weeks, while Jung and Park’s experimental group used an HMD for 30 min to study VRP and view the Chemoport insertion process in a 360 degree virtual angiography room [23,24].

### 4.2. Effectiveness of the Knowledge Test on Implantable Port-A Catheter

The study showed significant increases in post-intervention cognitive and delayed post-test knowledge scores, consistent with Tsai et al. [23] and Jung and Park [24]. These findings suggest that virtual reality is an effective cognitive learning strategy, enhancing knowledge retention through observation, simulated operation, and repetition in a virtual environment [37].

However, the study’s results differ from those of Ekstrand et al. [28], who found no significant difference in learning effectiveness between immersive VR simulations and traditional methods for learning neuroanatomy. This study’s focus on procedural knowledge in a contextual setting may explain this difference, as it more closely resembles real clinical situations and facilitates cognitive knowledge retention, while 3D neuroanatomy knowledge is more declarative and easier to forget.

### 4.3. Effectiveness of Implantable Port-A DOPS Assessment

The average DOPS assessment scores in the post-test and delayed post-test after the virtual reality teaching system intervention were significantly higher than the pre-test scores. This study’s skill learning outcomes resemble those of Maytin et al. [38], who designed a virtual reality scenario for Transvenous Lead Extraction (TLE) for doctors, finding that the experimental group outperformed the control group in practical skill operation, complication occurrence, and TLE operation time, with significant differences. One possible reason is that TLE, a high-risk, high-complication technique, cannot be practiced on real patients but can be improved through repeated practice and immediate feedback in virtual reality.

In contrast, this study’s delayed learning effect differs from that of Smith et al. [39], who used virtual reality simulation to train university nursing students in disaster decontamination skills. While the experimental group outperformed the control group in the post-test, both groups’ technical performance declined in the 5-month retention test, with the experimental group faring worse. One possible explanation is that disaster decontamination skills are less frequently applied in clinical practice due to the unpredictability of disasters, leading to reduced learning effectiveness. Additionally, the retention of knowledge and skills naturally declines over time and is gradually forgotten [40]. However, in this study, learners need to perform implantable Port-A catheter injection skills in clinical nursing practice, retaining the delayed learning effect in the delayed post-test.

### 4.4. Evaluation of Learning Attitude

The average learning attitude score towards the virtual reality learning system in this study was 4.29 ± 0.46. No comparable literature exists for analyzing the current learning attitude result, possibly because the virtual reality teaching incorporated animation, text explanations, and multiple feedbacks, creating realistic content and enhancing learners’ effective learning.

### 4.5. Learning System Satisfaction

This study revealed that nurses’ satisfaction with the virtual reality learning system had an overall average of 4.31 ± 0.58, with 39 participants (90.7%) feeling satisfied or very satisfied with the system’s overall learning effectiveness. The satisfaction results align with Tsai et al. [23] and Jung et al. [41]. These studies demonstrate that well-designed virtual reality teaching materials can help learners understand complex spatial relationships, increase motivation and confidence, and reduce learning fears, making virtual reality an effective learning tool. However, virtual reality cannot provide the same tactile experience as reality, making it difficult to perform accurate injections and affecting learners’ sense of reality. This study addressed this limitation by designing a picture-in-picture for the needle insertion process to be illustrated from a more precise angle, enabling learners to acquire the ability to correctly operate a Port-A catheter and enhancing their satisfaction with the learning system.

### 4.6. Learning Experience with Virtual Reality Teaching Materials

This study’s results showed that the average training time, level hints, and total errors decreased as the number of practice sessions increased, similar to Tsai et al. [23]. They used desktop virtual reality for Port-A injection skills intervention and found improvements in various aspects of technical practice. This analysis suggests that the VR simulations can enhance learners’ skill performance, indicating that through repeated practice or individualized teaching methods, VR reduces problems after skill mastery, effectively improving success rates, efficiency, and error reduction [37].

#### The Learning Experience of Virtual Reality Materials Using an Interview

Data from 10 learners’ experiences were analyzed and classified into five themes: (1) impressive realism of virtual reality scenarios, (2) helpfulness of virtual reality practice, (3) willingness to learn with virtual reality in the future, (4) limitations of virtual reality systems, and (5) potential for future curriculum planning. The theme of “limitations of virtual reality systems” was similar to the results of Lai et al.’s [18] study, which suggested increasing interactive functionality. However, this study’s learning system was a standardized process that did not allow learners to deviate or skip steps. It is recommended that future designs offer a more flexible learning system to enhance the learning experience.

The theme of “the helpfulness of practicing with virtual reality” in this study was similar to Yu et al.’s [35] study, which highlighted the benefits of endoscopic simulation training for medical students. Learners in this study consistently found virtual reality practice useful for enhancing their technical skills. The analysis suggests that traditional learning has focused mainly on knowledge, and a lack of practical experience has led to incomplete learning. Combining virtual reality teaching methods can help learners gain a comprehensive understanding of the course, making them more proficient in their future clinical operations.

### 4.7. Limitations

The limitations in this study include: (1) convenience sampling at a regional teaching hospital in North Taiwan with only 43 participants, limiting generalizability; (2) employing a quasi-experimental design with a single group but no control group for comparison; (3) focusing on nursing personnel with less than two years of work experience; the intervention was delayed by one month, resulting in two nursing personnel with 25 months of work experience; (4) room for improvement in the design of the virtual reality learning system; and (5) virtual reality lacking tactile experiences, affecting learners’ sense of reality.

## 5. Conclusions

This study employed a fully immersive and embedded Port-A catheter care scenario virtual reality teaching system, demonstrating positive learning effects. Diversified feedback methods helped learners integrate practice into clinical settings and interact with the teaching content, deepening their impression and arousing interest. Self-repeated learning produces positive reinforcement, strengthens behavioral responses, and results in more sustainable learning, improving effectiveness.

## Figures and Tables

**Figure 1 healthcare-11-01017-f001:**
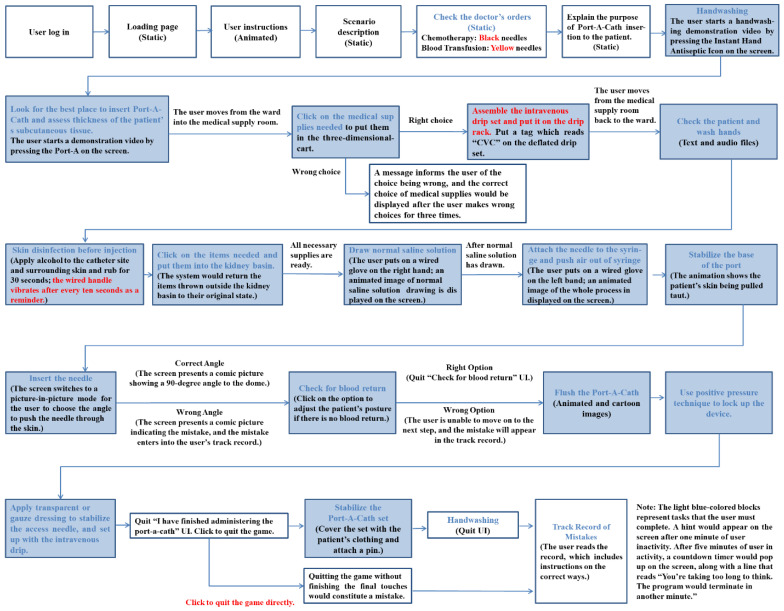
Procedure of a virtual reality implanted Port-A catheter. Note: Red color characters provide a hint for the user. Light blue background indicates main steps.

**Figure 2 healthcare-11-01017-f002:**
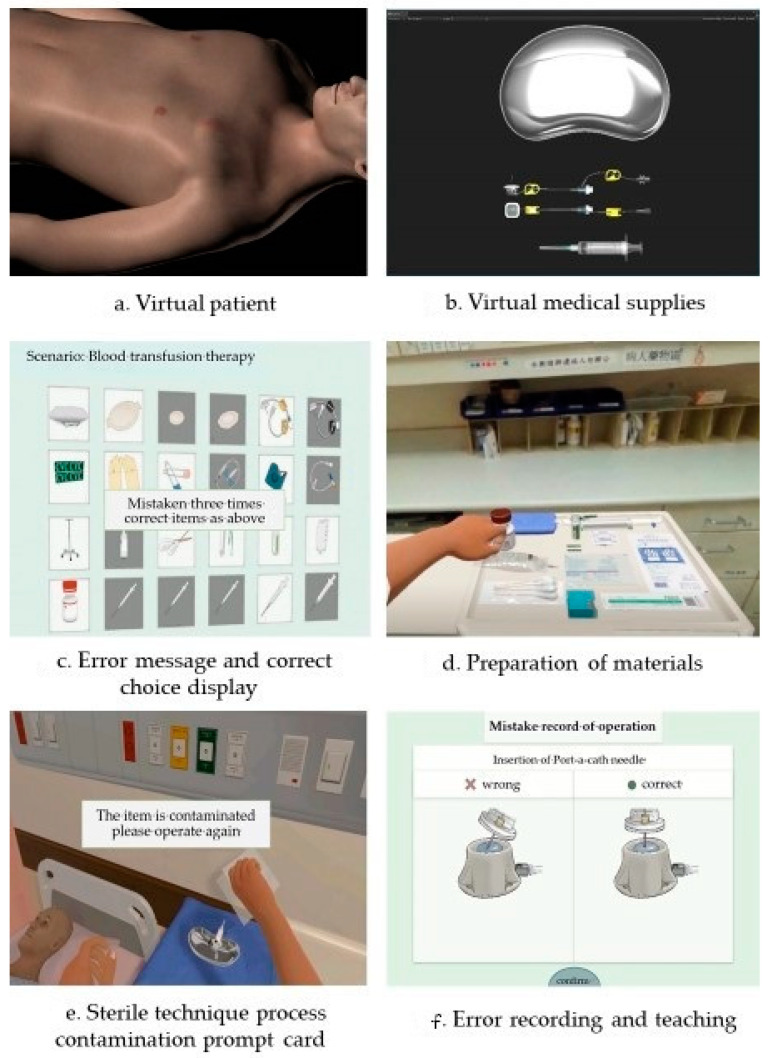
The learning process of the virtual reality teaching system.

**Figure 3 healthcare-11-01017-f003:**
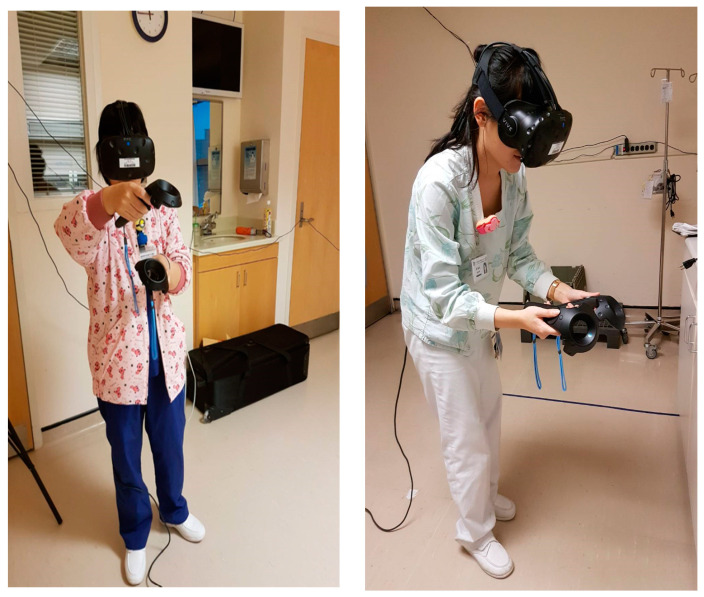
The learners’ actual operation process of the virtual reality learning system in two locations.

**Figure 4 healthcare-11-01017-f004:**
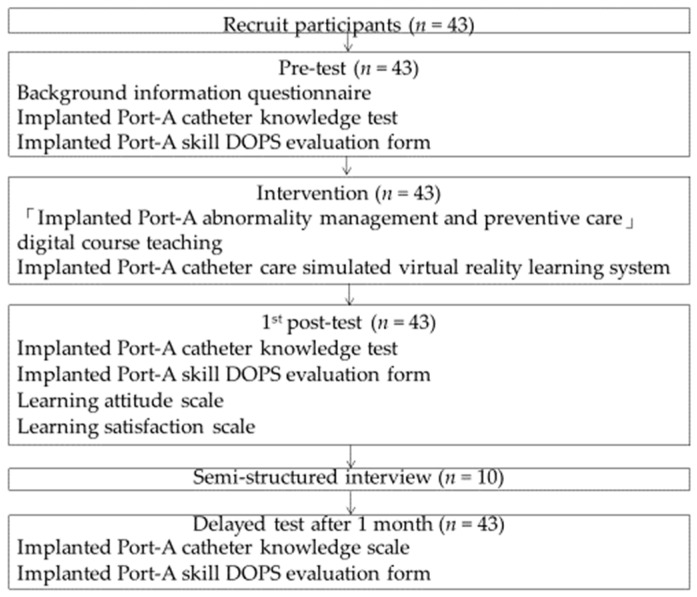
Procedure of the study.

**Table 1 healthcare-11-01017-t001:** Study design and measurements.

	O_1_	X	O_2_	O_3_
Measure	Two Weeks before the Intervention		Within 1 Week after the Intervention	One Month after the Intervention
Background information	√			
Knowledge of implanted Port-A catheter	√		√	√
DOPS evaluation of implanted Port-A catheter	√		√	√
Virtual reality teaching system learning attitude toward implanted Port-A catheter care			√	
Virtual reality teaching system satisfaction of implanted Port-A catheter care			√	

Note: O_1_: pre-test; O_2_: post-test; O_3_: delayed post-test; X: virtual reality teaching system intervention; and DOPS, Direct Observation of Procedural Skills; √ stands for measure time points.

**Table 2 healthcare-11-01017-t002:** Teaching design.

Item	Content
Theoretical framework	Situated learning theory
Duration	Around 1 h 10 min
Intervention	Implanted Port-A abnormality management, preventive care and Implanted Port-A catheter care simulated virtual reality operation instruction
Teaching strategies	1. TMS digital course teaching 1.1 Port-A care course (30 min) 1.2 Port-A abnormality management course (35 min) 2. Implanted Port-A catheter care virtual reality operation instruction (5 min)
Number of participants	43
Learning objectives	1. Can you really implement the Port-A catheter skill step and care procedure? 2. Distinguish the patient’s Port-A catheter care problems based on the patient’s signs and symptoms? 3. Select the correct clinical decision-making based on clinical Port-A catheter care problems?
Evaluation methods	Virtual reality teaching system knowledge scale for implanted Port-A catheter Evaluation form for implanted Port-A catheter DOPS Virtual reality teaching system for learning attitude scale for implanted Port-A catheter care Virtual reality teaching system satisfaction scale for implanted Port-A catheter care

**Table 3 healthcare-11-01017-t003:** Intervention.

**Unit**	**Number of learners** **(*n* = 43)**	**Pre-test**	**Intervention**		**Post-test**	**Interview** **(*n* = 10)**	**Delayed post-test**
		1. Background information 2. Knowledge test 3. DOPS	Digital course	VR practice	1. Knowledge test 2. Learning attitude 3. Satisfaction scale 4. DOPS		1. Knowledge test 2. DOPS
		**Date**	**Date of course**	**VR date of intervention**	**Date**	**Date**	**Date**
5N	8	6–10 Jan	11–15 Jan	15–19 Jan	12–16 Mar	19–23 Mar	16–20 Apr
5S	6			22–26 Jan			
6N	13			29 Jan–2 Feb			
4N	2			05–09 Feb			
6S	3			19–23 Feb			
4S	7			26 Feb–02 Mar			
ICU	4			05–09 Mar			
Total	43	5 days	5 days	5 days, respectively	5 days	7 days	5 days

Note: Jan, January; Feb, February, Mar, March, Apr, April.

**Table 4 healthcare-11-01017-t004:** Guidelines for virtual reality learning interviews on care for implantable Port-A catheters.

Interview Guidelines
Can you describe your learning experience and what you learned while practicing the VR simulation of Port-A catheter implantation?; How is practicing the VR simulation of Port-A catheter implantation different from your previous learning experiences? Did you have any profound experiences? Please provide an example; Does practicing the VR simulation of Port-A catheter implantation help with clinical skills for performing the procedure? Please explain; What difficulties did you encounter when using the VR teaching system, and how did you overcome them?; Do you believe that it is necessary for ongoing planning of the VR teaching systems? If you have the opportunity to use a VR teaching system again in the future, would you be willing to do so?; What are the pros and cons of using a VR teaching system for caring for patients with a Port-A catheter? Do you have any suggestions? Please describe it in detail.

**Table 5 healthcare-11-01017-t005:** Distribution of background information on nurses (*n* = 43).

Variable	*n* (%)	Range	Mean (SD)
Female	43 (100.0)		
Age in years		20–29	22.88 (1.68)
Education			
5 year diploma	3 (7.0)		
2 year college	24 (55.8)		
4 year college	7 (16.3)		
University	8 (18.6)		
Master’s	1 (2.3)		
On-the-job training			
No	41 (95.3)		
Yes	2 (4.7)		
Work experience in this hospital		3–25	9.86 (6.27)
Total nursing work experience		3–43	12.19 (9.60)
Nursing clinical ladder			
Probation	1 (2.3)		
N1	39 (90.7)		
N2	3 (7.0)		
Source of information about VR system			
None	14 (32.6)		
Yes (multiple choice)	29 (67.4)		
Newspaper	8 (18.6)		
Television	20 (46.5)		
Exhibition	5 (11.6)		
Others (supervisors, school, friends, and courses)	5 (11.6)		
VR experience			
None	30 (69.8)		
Yes (multiple choice)	13 (30.2)		
Participated in related exhibition	6 (14.0)		
Place for learning	1 (2.3)		
Other (entertainment, exhibition, movie, PS4, and games)	8 (18.6)		

**Table 6 healthcare-11-01017-t006:** Number of practitioners, total practice time, grand mean of level hint, and error frequency of thirteen times using virtual reality teaching materials for implanted Port-A catheter care.

Frequency	Number of Nurses	Total Practice Time	Grand Mean of Hint Frequency	Grand Mean of Error Frequency
T1	43	0:12:20	1.23	7.37
T2	43	0:09:52	0.79	6.40
T3	43	0:08:32	0.28	3.98
T4	40	0:07:36	0.10	2.85
T5	35	0:07:07	0.18	1.68
T6	15	0:06:46	0.27	1.60
T7	8	0:06:55	0.25	1.50
T8	6	0:06:59	0.00	0.83
T9	4	0:07:04	0.00	0.75
T10	3	0:06:44	0.00	0.67
T11	2	0:06:27	0.00	0.50
T12	1	0:05:49	0.00	1.00
T13	1	0:05:41	0.00	0.00

**Table 7 healthcare-11-01017-t007:** Distribution of learning attitude using virtual reality for implanted Port-A catheter care (*n* = 43).

		Disagree	Neutral	Agree	Mean	
No	Item	*n* (%)	*n* (%)	*n* (%)	(SD)	Rank
1	The virtual reality teaching system provides clear instructions on how to operate the system.	0 (0.0)	3 (7.0)	40 (93.0)	4.35 (0.61)	
2	The virtual reality teaching system is presented in an easy-to-understand way.	0 (0.0)	6 (14.0)	37 (86.0)	4.35 (0.72)	
3	The length of the content in the virtual reality teaching system is appropriate.	2 (4.7)	5 (11.6)	36 (83.7)	4.19 (0.82)	Low 3
4	The content of the virtual reality teaching system increases my interest in learning.	0 (0.0)	5 (11.6)	38 (88.4)	4.40 (0.70)	High 3
5	The virtual reality teaching system provides accurate information.	0 (0.0)	3 (7.0)	40 (93.0)	4.42 (0.63)	High 2
6	I can easily click on the content of the virtual reality teaching system.	1 (2.3)	3 (7.0)	39 (90.7)	4.23 (0.68)	
7	The virtual reality teaching system is a realistic and uncluttered scene that is easy to identify	0 (0.0)	2 (4.7)	41 (95.3)	4.49 (0.59)	High 1
8	I can experience the virtual reality teaching system without constantly moving or changing my posture.	0 (0.0)	4 (9.3)	39 (90.7)	4.23 (0.61)	
9	The use and operation of the virtual reality teaching system is helpful to me.	0 (0.0)	4 (9.3)	39 (90.7)	4.26 (0.62)	
10	I enjoy learning with virtual reality (VR) devices.	0 (0.0)	9 (20.9)	34 (79.1)	4.14 (0.74)	Low 1
11	I believe that this type of virtual reality (VR) learning is effective.	0 (0.0)	5 (11.6)	38 (88.4)	4.33 (0.68)	
12	In the future, I will be interested in learning using this method.	0 (0.0)	9 (20.9)	34 (79.1)	4.16 (0.75)	Low 2
	Grand mean of percent/mean (SD)	0.58	11.24	88.18	4.29 (0.46)	

## Data Availability

The data presented in this study are available on request to the authors.

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
