# Peer review of "Effects of Digital Learning and Virtual Reality in Port-A Catheter Training Course for Oncology Nurses: A Mixed-Methods Study"

_healthcare, 2023, doi:10.3390/healthcare11071017_

Round 1

Reviewer 1 Report

First of all, thank you for relying on me to review this manuscript. 

I would like to express my sincere congratulations for the research. The authors researched an interesting topic, the effects of digital learning and virtual reality on learning in the classroom. These have been  incorporated into training courses, and  are specifically interesting in the training of nursing techniques. 

The subject is relevant to the journal and it is interesting for readers.  

However, there are some aspects that I would like to comment on: 

The authors should check orthography, as well as update older references.  

Throughout the text the authors use port A-Cath and port A-catheter interchangeably, they should choose a single terminology and use the same terminology throughout the text. 

Title: The title reflects the subjects being investigated. Also, authors should consider adding the article type:  Mixed-methods.

Introduction: 

The introduction section is attractive to read from a reader point of view, explaining the background and reason for conducting this study. However, references 19 and 28 are very old. Authors should consider using more up-to-date references.

Materials and methods: 

2.1 Study Design 

The authors have used two research methodologies in their study, a quantitative and a qualitative methodology, something that is not clear in the study design. Authors should consider better differentiating the two methodologies and consider two separate methodological designs. 

The paragraph where the 3 studies that exist on VR are described should be deleted, and that information should be included in the discussion. 

2.2 Participants, Settings, and Procedure.  

Authors should explain why left-handed people have been excluded.

2.3.1 VR intervention of real-situated teaching system on implanted Port-A catheter care. 

To design the teaching system, the authors maintain that it went through multiple experts, but they do not mention how many there were. it would be advisable to better describe the design proccess, adding the number of experts. 

The authors also do not comment on how the scale was constructed. In addition, this was validated by only 3 experts. the authors should explain why only 3 experts participated in the validation of the scale, since it is an insufficient number. 

2.4.2 Evaluation form of Port A-DOPS.  

Why were items 4 and 7 removed from the original scale? 

2.4.3 Virtual reality teaching system knowledge scale of implanted Port-A catheter. 

The authors refer in line 295 that the original scale consisted of 22 items, later they report that they eliminated item 2 and 7, and that the final scale has 13 items. Where are the rest of the items that the original scale had? 

Item 2 and 7: what did they say, or what did they ask about. 

In line 296 the authors mention the S-CVI and I-CVI but do not report how many experts participated in the validation of the scale. 

2.3.2 Interventional procedures 

The authors have calculated a sample of 38 participants and have also taken losses into account. Why include 43 subjects in the study? 

2.5 Procedure  

The paragraph corresponding to the ethics committee, lines 331 to 333, is repeated in ethical considerations. The authors should consider removing it from this section, as it is reflected in the ethical considerations section. 

2.6 Data analysis 

The authors do not mention the types of post hoc tests used, but they do mention them in the results. They should also be mentioned in this section. 

The explanation of the data analysis of the qualitative part is very brief. 

3. Results

Table 5 is too large, taking into account that the columns on the right barely provide information. Should be configured differently. 

Authors should consider conveying the information contained in Table 8 in a single sentence, and delete the table. 

3.5. Use experience. 

The writing of the results of the interviews should be more objective, focused only on the results, leaving the evaluations for the discussion section. 

References 

The authors should review the references, and try to update them, since there are many references from more than 20 years ago. The topic you have discussed, virtual reality, is very active, and there are several recent studies that you can cite. 

Author Response

Point 1:The authors should check orthography, as well as update older references. Throughout the text the authors use port A-Cath and port A-catheter interchangeably, they should choose a single terminology and use the same terminology throughout the text.

Response 1: The references have been updated (pp.2-3). The term Port A-Cath has been changed to Port A-catheter throughout the text.

Point 2:

Title: The title reflects the subjects being investigated. Also, authors should consider adding the article type: Mixed-methods.

Response 2: The title now states it is a mixed-methods study (p.1, Line 3-4).

Point 3:

Introduction: The introduction section is attractive to read from a reader point of view, explaining the background and reason for conducting this study. However, references 19 and 28 are very old. Authors should consider using more up-to-date references.

Response 3: The introduction section has been revised. References 19, 29 (originally 28), and others have been updated (pp.1-3, Line 31-92、121-123).

Point 4:

Materials and methods:

2.1 Study Design: The authors have used two research methodologies in their study, a quantitative and a qualitative methodology, something that is not clear in the study design. Authors should consider better differentiating the two methodologies and consider two separate methodological designs.

The paragraph where the 3 studies that exist on VR are described should be deleted, and that information should be included in the discussion.

Response 4: The study design text has been revised (p.3, Line 95-110). Paragraph 2.3.1.1. 3 regarding existing VR studies has been deleted.

Point 5:

2.2. Participants, Settings, and Procedure: Authors should explain why left-handed people have been excluded

Response 5: The rationale for excluding left-handed people has been added the section of participants and setting (p.4, Line 118-121).

Point 6:

2.3.1. VR intervention of real-situated teaching system on implanted Port-A catheter care. 

To design the teaching system, the authors maintain that it went through multiple experts, but they do not mention how many there were. it would be advisable to better describe the design process, adding the number of experts. 

The authors also do not comment on how the scale was constructed. In addition, this was validated by only 3 experts. the authors should explain why only 3 experts participated in the validation of the scale, since it is an insufficient number.

Response 6: The VR design process involved 3 programmers and 5 nursing experts who provided their input on the nursing care and technical procedures in Koo Foundation Sun Yat-Sen Cancer Center and helped establish the entire Port-A catheter care procedure. Regarding the development of instruments, we invited 3 clinical nursing and education experts to validate the questionnaire because they are experts in the field and have extensive experience in Port-A catheter care procedures. The sentence has been revised (p.4, Line 130-131).

Lyn (1986) advised a minimum of 3 experts, but indicated that more than 10 was probably unnecessary (Pilot & Beck, 2006). In addition, Yusoff (2019) indicated “the minimum acceptable expert number is 2, however most of recommendations propose a minimum of 6 experts. Considering the recommendations (5–8) and the author’s experience, the number of experts for content validation should be at least 6 and does not exceed 10.”

Lynn, M. R. Determination and quantification of content validity. Nursing Research. 1986, 35, 382– 385.

Yusoff, M. S. B. ABC of content validation and content validity index calculation. Education in Medicine Journal. 2019, 11(2), 49–54. https://doi.org/10.21315/eimj2019.11.2.6

Point 7:

2.4.2. Evaluation form of Port A-DOPS: Why were items 4 and 7 removed from the original scale? 

Response 7: Items 4 and 7 were deleted due to low item usage frequency and corrected item-total correlation values < 0.30. This section has been revised (p.9, Line 225-227).

Point 8:

2.4.3. Virtual reality teaching system knowledge scale of implanted Port-A catheter: The authors refer in line 295 that the original scale consisted of 22 items, later they report that they eliminated item 2 and 7, and that the final scale has 13 items. Where are the rest of the items that the original scale had?

Item 2 and 7: what did they say, or what did they ask about.

Response 8: After being reviewed by five experts, two items were removed from the original 22-item knowledge scale based on expert opinions. One item was removed because its I-CVI was less than 0.80, and another item was removed because its answer could be correctly deduced from previous items. Furthermore, seven items were removed due to low discriminant validity. Thus, the final knowledge scale has 13 items, with an S-CVI of 0.85 and an I-CVI of 0.97. This section has been revised. (p.9, Line 237-241).

Point 9:

In line 296 the authors mention the S-CVI and I-CVI but do not report how many experts participated in the validation of the scale.

Response 9: Five experts participated in the validation of the satisfaction questionnaire. The sentence has been revised. (p.10, Line 267).

Point 10:

2.3.2 Interventional procedures: The authors have calculated a sample of 38 participants and have also taken losses into account. Why include 43 subjects in the study?

Response 10: The eligible participants were 43 after excluding 3 left-handed participants. Participants are interested in attending this study and have registered this study. Thus, we recruited all eligible participants after signed the consent form.

Point 11:

2.5 Procedure

The paragraph corresponding to the ethics committee, lines 331 to 333, is repeated in ethical considerations. The authors should consider removing it from this section, as it is reflected in the ethical considerations section.

Response 11: Original lines of 331 - 333 has been deleted (p.10, Line 277).

Point 12:

2.6 Data analysis

The authors do not mention the types of post hoc tests used, but they do mention them in the results. They should also be mentioned in this section.

The explanation of the data analysis of the qualitative part is very brief.

Response 12: Post hoc tests have been added to the sentence (p.11, Line 293-295), The data analysis of the qualitative part also has been added (p.11, Line 296-300).

Point 13:

  1. Results: Table 5 is too large, taking into account that the columns on the right barely provide information. Should be configured differently.

Authors should consider conveying the information contained in Table 8 in a single sentence, and delete the table.

Response 13: Table 5 has been reconfigured to reduce unnecessary variables (p.12, Line 317-318). A single sentence has been added to convey the information of interviewees (p.15, Line 384-385).

Point 14:

3.5. Use experience: The writing of the results of the interviews should be more objective, focused only on the results, leaving the evaluations for the discussion section.

Response 14:

The results of the interviews have been revised (pp.15-16, Line 391、405、415、424、434).

Point 15:

References: The authors should review the references, and try to update them, since there are many references from more than 20 years ago. The topic you have discussed, virtual reality, is very active, and there are several recent studies that you can cite.

Response 15: References have been reviewed and updated (pp.1-3, Line 572).

Reviewer 2 Report

The proposed work is quite interesting however the are few things that draw my attention:

1.  Table 1 shows that 2 week before intervention, within 1 week after intervention and 1 month after intervention. From where you decided to use this duration? why one week? why not 2 days? or maybe 2 weeks? or ten days? do you have any reference that support this duration?

2. The experiment shows the comparison between pre test and post test? however i didnt get any comparison with similar work.

3. Table 2 teaching design, the learning process require 1 hour and 10 minute. Is this average of period?.  

Author Response

Point 1: Table 1 shows that 2 week before intervention, within 1 week after intervention and 1 month after intervention. From where you decided to use this duration? why one week? why not 2 days? or maybe 2 weeks? or ten days? do you have any reference that support this duration?

Response 1: This study referred to relevant literature such as Tsai et al. [23], who used three weeks after the intervention as the delayed test time point. Other studies with similar designs were also consulted, such as Ekstrand et al. [28], who used seven days after the intervention, and Smith et al. [39], who used 5 months after the intervention as the delayed test time point. The timing of delayed tests in this study was determined by considering factors such as the number of participants, clinical practice, and time constraints. (p.3, Line 101-103、p.17, Line 475-479).

Point 2: The experiment shows the comparison between pretest and posttest? However I didn’t get any comparison with similar work.

Response 2: Regarding the comparison with similar work, the study added relevant research and comparison content in the Discussion section (p.17, Line 447-484).

Point 3: Table 2 teaching design, the learning process require 1 hour and 10 minutes. Is this average of period?

Response 3: The learning process requires 1 hour and 10 minutes (Table 2). It is the total learning time required for the digital course and is not an average time period. Detailed time spent has been added to Table 2 (pp.7-8, Line 191-192).

Reviewer 3 Report

General comments:

Thank you for inviting me to review the article "Effects of Digital Learning and Virtual Reality in Port-A Catheter Training Course for Oncology Nurses." The present study is interesting. However, I suggest modifying the article according to the comments.

Title:

  1. It is essential to mention the study design in the abstract.

Abstract:

  1. With the restricted 200 words count (according to healthcare, MDPI policy) in the abstract, the authors must prioritize methods, results, and essential conclusions. However, the authors emphasized more about the background. Kindly include the significant level (p-value) and strength of association (if available) in the abstract.

Introduction:

  1. Fairly written. However, the authors need to be more accurate and to the point of bringing the introduction to the rationale. But I consider the introduction words are too much for a research article. Hence, I suggest reducing the introduction content.
  2. 1.2: Program design. It needs to be shifted to the methods section.

Methods:

  1. Please justify the reasons for the suggested gap in assessing the pre-test and post-test gap duration.
  2. The standardization of the training program needs more explanation.
  3. Even though the authors mentioned the Cronbach alpha value of the questionnaire (line 321), the authors need to explain more psychometric properties of the tool used.

Results:

  1. Results are well presented, but analysis regarding the open-ended questions needs to be explained.

Discussion:

  1. The authors' discussion about the results is too much limited. Even though the authors mentioned that there were limited studies in this context, it is essential to update the works of literature in the discussion (as the study was conducted almost five years before, and a lot of changes happened).
  2. Authors must link the present study findings and its implication in the current scenario to make the study more imperative.

Author Response

Point 1: Title: It is essential to mention the study design in the abstract.

Response 1: The study design in the abstract has been revised as a mixed-methods research design (p.1, Line 16).

Point 2: Abstract: With the restricted 200 words count (according to healthcare, MDPI policy) in the abstract, the authors must prioritize methods, results, and essential conclusions. However, the authors emphasized more about the background. Kindly include the significant level (p-value) and strength of association (if available) in the abstract.

Response 2: The content of the abstract has been revised (p.1, Line 14-25).

Point 3: Introduction: Fairly written. However, the authors need to be more accurate and to the point of bringing the introduction to the rationale. But I consider the introduction words are too much for a research article. Hence, I suggest reducing the introduction content.

1.2: Program design. It needs to be shifted to the methods section.

Response 3: The section of the introduction has been revised (pp.1-3, Line 31-92).

Point 4: Methods: Please justify the reasons for the suggested gap in assessing the pre-test and post-test gap duration.

Response 4: This study referred to relevant literature such as Tsai et al. [23], who used three weeks after the intervention as the delayed test time point. Other studies with similar designs were also consulted, such as Ekstrand et al. [28], who used seven days after the intervention, and Smith et al. [39], who used 5 months after the intervention as the delayed test time point. The timing of delayed tests in this study was determined by considering factors such as the number of participants, clinical practice, and time constraints. (p.3, Line 101-103、p.17, Line 475-479).

Point 5: Methods: The standardization of the training program needs more explanation.

Response 5: The standardization of the training program has been added. (p.5, Line 165-171)

Point 6: Methods: Even though the authors mentioned the Cronbach alpha value of the questionnaire (line 321), the authors need to explain more psychometric properties of the tool used.

Response 6: The psychometric properties of the tools have been added (p.9, Line 227-230, & 246-247; p.10, Line 257-259, 268-270).

Point 7: Results: Results are well presented, but analysis regarding the open-ended questions needs to be explained.

Response 7: Qualitative feedback on two open-ended questions are presented on p.15, Line 379-382 and have interpreted on p.18, Line 495-503

Point 8:

Discussion:

  1. The authors' discussion about the results is too much limited. Even though the authors mentioned that there were limited studies in this context, it is essential to update the works of literature in the discussion (as the study was conducted almost five years before, and a lot of changes happened).
  2. Authors must link the present study findings and its implication in the current scenario to make the study more imperative.

Response 8: Regarding the comparison with similar work, the study has added relevant research and comparison content in the discussion section. (p.17, Line 468-484).

Round 2

Reviewer 1 Report

The authors have done a good job, modifying according to the proposed suggestions.

Reviewer 3 Report

Dear authors,

Thanks for making necessary changes. Great work indeed!

Wish you all the best.